# Quality Improvement of Green Saba Banana Flour Steamed Cake

**Jau-Shya Lee** [1,*] **, NurDiyana Yusoff** [2] **, Ai Ling Ho** [1] **, Chee Kiong Siew** [1] **, Jahurul Haque Akanda** [3] **and Wan Xin Tan** [1]

1 Faculty of Food Science and Nutrition, University Malaysia Sabah, Jalan UMS, Kota Kinabalu 88400, Sabah, Malaysia
2 Agriculture Research Centre, Department of Agriculture Sabah, Tuaran 89207, Sabah, Malaysia
3 Department of Agriculture, School of Agriculture, University of Arkansas, 1200 North University Drive, M/S 4913, Pine Bluff, AR 71601, USA
* Correspondence: jslee@ums.edu.my

**Abstract:** Gluten avoidance is becoming a popular diet trend around the world. In this study, green Saba banana flour (GSBF) was used to produce a gluten-free (GF) steamed cake. The effects of soy protein isolate (SPI) (0%, 10%, 15%) and Ovalette (0%, 3.5%, 7%) on the quality of the cake were investigated. Physicochemical properties of the flours were measured. The viscosity and specific gravity of the batters; as well as the specific volume, weight loss and texture profile of the resulting cakes were determined. Sensory evaluation was performed to compare the acceptance of the cake formulations. The macronutrient and resistant starch content of the cakes were determined. The use of an appropriate level of SPI and Ovalette was found to effectively enhance the aeration of the cake batter and improved the specific volume and weight loss of the cake. The presence of Ovalette was essential to soften the texture of the cake. GF cake supplemented with 10% SPI and 3.5% Ovalette obtained the highest sensorial acceptance. The nutritional quality of this sample was significantly improved, whereby it contained higher protein than the gluten-containing counterpart. GSBF also contributed to the high dietary fiber and resistant starch content of the cake.

**Keywords:** green banana flour; soy protein isolate; emulsifier; cake; gluten-free; resistant starch; dietary fiber

## 1. Introduction

Celiac Disease (CD) is a major public health problem worldwide, the prevalence of this disease is 1.4% based on serologic test results [1]. It is an autoimmune digestive disorder caused by ingestion of gluten (particularly gliadin peptides) that leads to injury of the small bowel of the patients, subsequently causing nutritional deficiency, development of fertility-related complications and malignancy [2,3]. The prevalence of CD is believed to be heavily underestimated due to the frequently misdiagnose with other irritable bowel syndromes and a lack of awareness among medical professionals about the extra-intestinal presentations of the disease [4,5]. Till now, the only effective treatment available to CD patients is a stringent lifetime gluten-free (GF) diet [6]. In addition to the needs of CD patients, GF foods are also in demand by consumers with gluten sensitivity or non-coeliac gluten intolerance [7]. In recent years, GF foods have also become popular among consumers without CD [8]. This has resulted in an upsurge in demand and drastic growth in the global GF food market size.

Gluten refers to a group of storage proteins made up of gliadins and glutenins. Hydrated glutenins are more cohesive and contribute to the dough strength and elasticity, whereas hydrated gliadins behave as a plasticizer for glutenins and are responsible for the viscosity and extensibility of the dough system [9]. Avoidance of gluten in bakery products by switching from using conventional wheat flour to other low protein flours indicates

a loss of technological quality and nutritional quality of the final products. As a result, it is of paramount importance to replace proteins from other sources in the GF product formulations to achieve the desirable properties such as to enhance the Maillard browning and flavor formation while cooking; to improve the viscoelasticity of the dough or batter; as well as to improve the structure of the final products by aiding the gelation and foaming process [10].

Green banana flour is known to contain high resistant starch (RS) which is resistant to digestion. Intake of RS has been reported as able to reduce postprandial insulinemic and elevated glycemic responses. Other metabolic health-associated benefits of RS are increasing satiety, reducing fat storage, improving insulin sensitivity, lowering triglyceride and plasma cholesterol concentrations [11,12]. Through a proper modulation of SCFAs in the human body, the obesity-related metabolic disorders, and their associated diseases, such as type 2 diabetes, hypertension can be prevented [13,14]. These attractive health benefits have called attention to the development of various food products using green banana flour. Segundo et al. (2017) [15] found that the substitution of wheat flour with green banana flour increased the dietary fiber, resistant starch, polyphenol content and antioxidant activity of both layer cakes and sponge cakes. They also found that the particle size of green banana flour affected the nutritional quality of the cakes, where coarse flours yielded cakes with a higher content of dietary fiber, and fine flours yielded cakes with a significantly higher RS content. Another study also disclosed that the fiber content and the antioxidant properties of cake samples were enhanced upon addition of banana flour from 2% to 10%, however, a negative impact on textural and sensory profile was noticed beyond 8% of incorporation [16].

Soy Protein Isolate (SPI) is a highly purified protein that is isolated from soybean. It has good gelling, emulsifying and foaming properties and is often used as a functional ingredient in food industry. SPI was used to replace the protein in GF food such as rice-cassava bread [10] and banana-cassava pasta [17]. SPI could effectively improve the total phenolic content, antioxidant capacities, enhance amino acid profiles and increase the protein digestibility of gluten-free pasta made of mixture of banana flour and cassava flour [17].

Emulsifiers are key ingredients for the successful production of bakery products. They are surface-active, and amphiphilic by nature. They typically improve the aeration and fine dispersion of air bubbles in the batter or dough system. They also have crumb-softening and anti-staling effects and can help improve cake volume [18]. The straight fatty acid-chain of emulsifier molecules can form a complex with the helical structure of amylose in starch, thus reducing the rate of starch retrogradation [18]. Several authors have incorporated emulsifiers in gluten-free formulations to improve the quality of the products [19–22].

Saba banana (ABB triploid hybrid) is a cooking banana most abundantly found in the state of Sabah, Malaysia. The banana is a commercial crop that has been distributed to other parts of Malaysia, and Brunei Darussalam since 2010 [23]. It is commonly consumed as fried banana fritters, steamed, or used to make other local snacks. The downstream industrial applications are still limited. The unripe green Saba banana flour (GSBF) is reported to contain high RS with an estimated glycemic index (GI) of 47.48 [24]; thus, can be classified as a low GI food [25]. With its low digestibility and low cost, green Saba banana flour is an excellent alternative for development of GF foods to increase the food choice for individuals on a GF diet. However, green banana flour lacks the nutritive value, particularly protein, and technological functionality of wheat flour, leading a need to supplement the functional ingredients in the product formulation. This project was aimed at developing a gluten-free cake using GSBF, SPI and a commercial emulsifier, Ovalette. The objective of this study was to characterize the batter and cake quality upon addition of SPI and Ovalette, and subsequently the sensory acceptance and nutritional quality of the final product.

## 2. Materials and Methods

### 2.1. Materials

Matured Saba banana (*Musa acuminata* × *Musa balbisiana*) was collected from an orchard in Keningau, Sabah. Banana fruits without physical defects and with total green color peel (matured but unripe) were immediately processed into flour upon arrival in the laboratory. Soy protein isolate (SPI) with a protein content of 88% was purchased from Thong Sheng Food Technology Sdn. Bhd., Pulau Pinang, Malaysia, Ovalette, a commercial gel phase emulsifier mixture of mono- and diglycerides and polyglycerol esters of fatty acids was bought from Bakels (Malaysia) Sdn Bhd. (Shah Alam, Malaysia). Wheat flour (Cap Sauh, Johor Bahru Flour Bill Sdn. Bhd., Johor, Malaysia), castor sugar (Gula Prai; Malayan Sugar Manufacturing Corporation Bhd., Kuala Lumpur, Malaysia), full cream milk powder (Dutch Lady; Dutch Lady Milk Industries Bhd., Petaling Jaya, Malaysia), palm olein (VeSawit; YL Brands Sdn. Bhd., Kuala Lumpur, Malaysia), baking powder, salt and vanilla essence were locally purchased. Other chemicals were used as received without further purification.

### 2.2. Preparation of Green Saba Banana Flour

The method of Lee et al. [24] was followed to prepare the green Saba banana flour (GSBF). The freshly received green banana was peeled, sliced (2 mm thick), and immediately soaked in 0.5% (*w/v*) citric acid for 10 min. The banana slices were then dried at 50 °C for 24 h. The dried chips were ground and sieved through 60-mesh screen. The flour was kept in an air-tight container until further use.

### 2.3. Preparation of Cake

The basic formulation of the cake (based on flour weight) consisted of 100% flour, 100% sugar, 100% egg, 20% full-cream milk powder, 15% palm olein, 4% vanilla essence, 2% baking powder and 1% salt. Two independent variables were the SPI (soy protein isolate) and Ovalette; which were tested at three levels (SPI: 0%, 10% and 15%; Ovalette: 0%, 3.5% and 7%) in the GSBF cakes. The levels of these additives were determined from the preliminary trials in the laboratory. A total of eight cake formulations were investigated. On top of that, two additional cake formulations were also prepared for comparison, namely the standard and the control. The standard (positive reference for comparison) was the gluten-containing cake made of 100% wheat flour; meanwhile, the cake made of 100% GSBF without SPI and Ovalette was used as the control of the experiment. Table 1 shows the cake formulations that underwent comparison.

**Table 1.** The cake formulations under investigation.

| Sample | Ingredient (%) * | | | |
|---|---|---|---|---|
| | Wheat Flour | GSBF | Ovalette | SPI |
| Standard | 100 | 0 | 0 | 0 |
| Control | 0 | 100 | 0 | 0 |
| O0P10 | 0 | 100 | 0 | 10 |
| O0P15 | 0 | 100 | 0 | 15 |
| O35P0 | 0 | 100 | 3.5 | 0 |
| O35P10 | 0 | 100 | 3.5 | 10 |
| O35P15 | 0 | 100 | 3.5 | 15 |
| O7P0 | 0 | 100 | 7 | 0 |
| O7P10 | 0 | 100 | 7 | 10 |
| O7P15 | 0 | 100 | 7 | 15 |

* The % of the ingredient was based on the % of flour. GSBF—green Saba banana flour; SPI—soy protein isolate.

The steam cake was prepared according to the method of Itthivadhanapong et al. [26] with slight modifications. The egg, sugar, salt and Ovalette were mixed in a cake mixer (KitchenAid 5KsM150 Stand Mixer, KitchenAid, Benton Harbor, MI, USA) at speed No.

5 for 5 min until the batter turned white and became fluffy. After that, the flour, milk powder, SPI and palm olein were added with continuous stirring at speed No. 3 (2 min). After mixing, the batter was poured into a rectangular cooking pan (18 cm length × 9 cm width × 6 cm height) and steamed in a preheated steamer (15 min preheating) for 30 min. After steaming, the cake was removed from the pan and allowed to cool down for 1 h at ambient temperature. The cake was kept in an air-tight container at room temperature (25 °C) prior to further analysis.

### 2.4. Measurement of Color

The color of the flour was determined by a Minolta colorimeter (Konica CR 400; Osaka, Japan). Results were expressed in the CIE *L*a*b** color space using the D65 standard illuminant and the 10° standard observer. The *L** coordinate is a measure of lightness, with 0 being black and 100 representing white. The *a** coordinate represents the green to red color range, and a positive *a** value indicates redness. The *b** coordinate represents the blue to yellow color range, and a positive *b** value indicates yellowness. Five g of flour sample was firmly pressed into a glass petri dish (outer diameter of 5 cm), and the surface of the flour was leveled before the measurement was taken.

### 2.5. Water Holding Capacity

The water-holding capacity of the flour was determined according to Mesías and Morales [27] with slight modifications. One g of flour and 25 mL of distilled water were added into a pre-weighed centrifuged tube and vigorously vortexed for 1 min. The tube was held at room temperature for 30 min prior to centrifugation at $3000 \times g$ for 20 min. The supernatant was discarded, and the tube was weighed. The water-holding capacity was calculated by the following formula:

$$\text{Water holding capacity} = \frac{W2 - W1}{W0} \times 100 \tag{1}$$

where
    W0 = weight of flour;
    W1 = weight of centrifuge tube with flour;
    W2 = weight of centrifuge tube with sediment.

### 2.6. Oil Holding Capacity

One g of flour and twenty-five mL of palm oil were added into a pre-weighed centrifuged tube and the content was mixed using a vortex mixer for 2 min. The tube was allowed to stand at room temperature for 30 min prior to centrifugation at $3000 \times g$ for 20 min. The supernatant was discarded, and the tube was weighed. Oil-holding capacity was calculated by the following formula:

$$\text{Oil holding capacity} = \frac{W2 - W1}{W0} \times 100 \tag{2}$$

where
    W0 = weight of flour;
    W1 = weight of centrifuge tube with flour;
    W2 = weight of centrifuge tube with sediment.

### 2.7. Proximate Analysis

The proximate compositions were determined in accordance with AOAC methods [28]. The moisture content of the sample was determined after drying 3 g of sample in a 105 °C oven for 24 h (AOAC 931.01). The loss of weight was recorded as the moisture content. For protein content determination, the Kjeldahl method (AOAC 2001.11) was used, whereby 3 g of sample was first hydrolyzed with 15 mL of concentrated sulfuric acid containing two copper catalyst tablets in a heat block at 400 °C for 2 h. After that, the nitrogen content of the

digested sample was determined by the Kjeldahl analyzer (Kjeltec 2300; FOSS, Hilleroed, Denmark). The total nitrogen content was then multiplied by 6.25 to estimate the total protein content of sample. The fat content was determined by Soxhlet extraction method. The sample (3 g) was extracted with 90 mL of petroleum ether for 1 h 20 min in a Soxhlet extraction system (Soxtec 8000; FOSS, Hilleroed, Denmark) (AOAC 991.36). The ash content of the sample was determined by ashing 3 g of sample in a muffle furnace (Carbolite, Hope Valley, UK) set at 550 °C for 24 h (AOAC 930.05).

### 2.8. Determination of Total Dietary Fiber

The total dietary fiber content of sample was determined following the procedures provided by Megazyme TDF test kit (K-TDFR; Megazymes, Wicklow, Ireland). Briefly, 1 g of sample was subjected to sequential enzymatic digestion by heat-stable α-amylase, protease, and amyloglucosidase. After enzymatic hydrolysis, pre-heated ethanol (60 °C, 95%) was added to precipitate the dietary fiber in the sample. The precipitation process take place at room temperature for 60 min. The precipitated dietary fiber was recovered by filtering the solution through a celite-filled crucible fitted to Fibertec 1023 (FOSS, Denmark). Then, the crucible was dry overnight in a 105 °C oven. The dried residue was then measured for ash and protein content. Total dietary fiber is the weight of the filtered and dried residue after deducting the weight of protein and ash.

### 2.9. Determination of Resistant Starch

The Resistant Starch (RS) content was determined using Megazyme RS Kit (K-RSTAR; Megazyme International Ireland Ltd., Co., Wicklow, Ireland). About 100 mg of sample was added with pancreatic α-amylase (10 mg/mL) and amyloglucosidase (3 U/mL), followed by incubation at 37 °C for 16 h under continuous agitation (WB14; Memmert, Schwabach, Germany). Four mL of ethanol (99% *v/v*) was added to terminate the reaction, the RS was recovered as a pellet on centrifugation (3000 rpm for 10 min). The supernatant was decanted and re-suspended in 2 mL of 50% ethanol with vigorous mixing. Six mL of ethanol was further added and subjected to centrifugation at 3000 rpm for another 10 min. The ethanol suspension and centrifugation steps were repeated twice, prior to dissolving the RS pellet in 2 M KOH by vigorous stirring in an ice-water bath. The solution was neutralized with acetate buffer and hydrolyzed with amyloglucosidase (0.1 mL, 3300 U/mL) for 20 min at 50 °C. The solution was then transferred to a 100 mL volumetric flask and adjusted to 100 mL with distilled water and mixed well. 0.1 mL of aliquots was then diluted with 3 mL of Glucose Determination Reagent (GOPOD Reagent) and incubated at 50 °C for 20 min. The absorbance was measured at 510 nm (Lambda 35; Perkin-Elmer, Buckinghamshire, UK) against the reagent blank. The resistant starch content in the test samples was calculated as follows:

$$\text{Resistant starch (g/100 g sample)} = \Delta E \times F \times 100/0.1 \times 1/1000 \times 100/W \times 162/180 \tag{3}$$

where
$\Delta E$ = absorbance read against the reagent blank;
F = conversion from absorbance to micrograms;
100/0.1 = volume correction (0.1 mL taken from 100 mL);
1/1000 = conversion from micrograms to milligrams;
W = dry weight of sample;
162/180 = factor to convert from free D-glucose, as determined, to anhydro-D-glucose as occurs in starch.

### 2.10. Batter Specific Gravity

Batter specific gravity was calculated by dividing the weight of batter over the weight of an equal volume of the water [29]. The specific gravity of the batter was defined as the weight of the batter against the weight of the water.

### 2.11. Batter Viscosity

The cake batter viscosity was measured using a viscometer (DV-II+ Viscometer; Brookfield, WI, USA). The measurement was carried out after the batter was rested for 10 min after mixing was completed. Two hundred ml of batter was placed into a 500 mL beaker up to a level marked near the brim. Spindle No. 07 and test speed of 30 rpm were used to determine the viscosity at room temperature (25 °C). The viscosity value was recorded after 1 min of shearing.

### 2.12. Texture Profile Analysis

Texture Profile Analysis (TPA) of the samples was conducted using a TA.XTPlus Texture Analyzer (TA-XT Plus; Stable Micro Systems Ltd., Godalming, Surrey, UK) with as 80 mm aluminum cylindrical probe. The cake samples were cut into $2 \times 2 \times 2$ cm$^3$ cubes (with the crust removed) and subjected to a programmed double-cycle compression and the texture profile was determined using Texture Expert 1.05 software (Stable Microsystems). The crumb was compressed to 25% of its initial height at 2 mm/s. Thirty s delay was set between the first and second compression. The hardness, springiness, chewiness and resilience were obtained from the force–time curve of the texture profile. The texture parameter of cake was averaged from 10 sub-samples of two replicates (total 20 measurements).

### 2.13. Sensory Evaluation

Sensory evaluation was conducted on the day after the cakes were prepared in individual booths with cool, natural, fluorescent light. The tests were carried out at the Laboratory of Sensory Evaluation located at the Faculty of Food Science and Nutrition, Universiti Malaysia Sabah. Because of the high number of cake formulations, two types of sensory tests were performed: Ranking Test and Hedonic Test. Ranking Test using Balanced Incomplete Block (BIB) design was first carried out to discriminate the least preferred formulations (total of eight formulations, $t = 8$) to avoid the potential sensory fatigue among the panelists [30] in the subsequent Hedonic Test. In BIB design (Table S1 (Supplementary Materials)), every formulation was replicated seven times ($r = 7$) and all pair of cake formulations occurred three times ($\lambda = 3$) in 14 blocks ($b = 14$) [31]. A total of 42 untrained healthy panelists ranked their preference over 4 samples ($k = 4$) randomly assigned to them (1 = like the most and 4 = dislike the most).

Forty healthy panelists recruited from the students and staff of the faculty were involved in the Hedonic test. They were asked to evaluate the volume, color, aroma, taste, softness, moistness, and overall acceptability of the cake samples assigned. A typical nine-point hedonic scale by Jones et al. [32] was used (1 = dislike extremely, 5 = neither like nor dislike and 9 = like extremely).

In both sensory evaluation sessions, cake slices (approximately 10 g each; equivalent to two-bite portions) coded with three-digit numbers were served along with drinking water for palate cleansing. The panelists were advised to rinse their mouth in between each sample testing. All the cake samples were halved (cross-sectional) and presented to the panelists for the evaluation of the cake volume. The panelists were asked to observe and compare the volume of the cakes prior to evaluation of other attributes.

### 2.14. Statistical Analysis

The results were the average of at least triplicate measurements except for the data of TPA and Sensory Evaluation. The data were analyzed using one-way analysis of variance (ANOVA) with SPSS ver. 24 (Statistical Package for Social Science). The means were compared at 95% confidence interval. Non-parametric data from Ranking Test were analyzed with Friedman's test. Fisher's Least Significant Difference (*LSD*) test was conducted to determine the difference between the samples when the null hypothesis of Friedman's Test was rejected. The Kruskal–Wallis test and Mann–Whitney U test were employed to examine the statistical difference for the sensory attributes tested in Hedonic Test.

## 3. Results and Discussion

### 3.1. Flour Analysis

Several basic chemical compositions of green Saba banana flour (GSBF) were determined and compared to commercial wheat flour (WF) (Table 2). As expected, WF contains higher protein and fat, whereas SGBF contains higher ash, dietary fiber and resistant starch (RS) ($p < 0.05$). The protein, fat, ash, and RS content of GSBF agrees with the earlier reported values [33,34]. RS in green banana flour is the non-digestible polysaccharides that behaves in the same way as dietary fiber. It is resistant to digestion but can be fermented by colonic microbiota to produce short chain fatty acids with positive metabolic effects [35,36]; additionally, it has the ability to prevent obesity, type 2 diabetes and hypertension [14]. The dietary fiber in banana flour consists of mixture of soluble fraction (pectin) and insoluble fraction (cellulose, lignin and hemicellulose) [37]. Other commonly used gluten-free flours such as maize and rice have also been reported to have lower levels of fiber as compared to WF [38,39]; thus, have caused the resultant products to have lower nutritional quality.

**Table 2.** Chemical compositions of commercial wheat flour and green Saba banana flour.

| Composition | Wheat Flour (WF) | Banana Flour (GSBF) |
|---|---|---|
| Moisture (%) | 13.44 ± 0.37 [b] | 11.81 ± 0.24 [a] |
| Protein * (%) | 11.69 ± 0.05 [b] | 3.87 ± 0.02 [a] |
| Fat * (%) | 0.91 ± 0.01 [b] | 0.41 ± 0.01 [a] |
| Ash * (%) | 0.85 ± 0.03 [a] | 1.96 ± 0.02 [b] |
| Dietary fiber * (%) | 2.82 ± 0.02 [a] | 10.22 ± 0.16 [b] |
| Resistant starch * (%) | 27.9 ± 0.27 [a] | 68.9 ± 0.14 [b] |

* Dry matter basis. Means with identical alphabet within the same row indicate insignificant difference ($p > 0.05$).

In raw form, the RS in WF and GSBF was inherently RS2 that was protected from digestion by the crystalline structure of the starch granules. GSBF contained more highly dense starch granules [40] which were more difficult for the digestive enzyme to penetrate, hence a higher RS than in WF.

Apart from basic chemical composition, the color, water and oil-holding capacity of WF and GSBF were also compared (Table 3). The color of the flour is expressed using three values: $L*$ value ranges from 0 to 100, where 0 illustrates black and 100 represents white, $a*$ value corresponds to green ($-$) and red ($+$), whereas $b*$ corresponds to blue ($-$) and yellow ($+$). GSBF was found to be less white, darker with more intense redness and yellowness than WF ($p < 0.05$). A commercial baking white wheat flour was used in this study, typically, this flour is bleached to remove the yellow pigment (xanthophyll) to make it whiter. Besides not being treated with bleaching agent, the yellowish color of GSBF was also contributed by the tissue browning by enzymatic oxidation of the phenolic compounds in the fruit.

**Table 3.** Color, water-holding capacity and oil-holding capacity of wheat flour (WF) and green Saba banana flour (GSBF).

| Characteristic | Wheat Flour (WF) | Banana Flour (GSBF) |
|---|---|---|
| Color | | |
| Brightness ($L*$) | 90.74 ± 0.13 [b] | 82.72 ± 0.01 [a] |
| Redness ($a*$) | 0.30 ± 0.02 [a] | 1.46 ± 0.04 [b] |
| Yellowness ($b*$) | 8.95 ± 0.04 [a] | 9.62 ± 0.04 [b] |
| Water-holding capacity (%) | 74.67 ± 3.51 [a] | 172.00 ± 4.03 [b] |
| Oil-holding capacity (%) | 85.67 ± 2.08 [b] | 64.33 ± 1.53 [a] |

Means with identical alphabet within the same row indicate insignificant difference ($p > 0.05$).

### 3.2. Batter Analysis

The viscosity and specific gravity of the cake batters were measured. Figure 1 shows that the batter prepared using 100% GSBF reported the lowest viscosity and the highest specific gravity ($p < 0.05$) among all tested samples. Gluten in the standard batter (100% WF) provided the desirable, unique rheological properties of the batter by conferring water absorption capacity, cohesivity, viscosity and elasticity [9]. Higher viscosity in the WF batter was attributed to the gluten development [41] after hydration by water and mixing. In the absence of gluten, the control exhibited high fluidity; little air could be incorporated in the batter during mixing, which led to a high batter density. The addition of SPI and Ovalette (O0P10, O0P15, O35P0 and O7P0) significantly increased the batter viscosity and improved the air retention of the batter ($p < 0.05$). SPI was reported to have higher water-adsorption capacity as compared to wheat protein [42] and resulted in an increment of the gluten-free batter viscosity as reported earlier [43]. Compared to the batter formulations containing single additive (O0P10 and O0P15 vs. O35P0 and O7P0), the combination of two additives would further reduce the specific gravity of the batters (Figure 1). Since the introduction of SPI and Ovalette to GSBF was based on an addition basis, the use of 10% and 15% of SPI elevated the total solid content of the formulations and thus their viscosities. It is noteworthy that the presence of Ovalette in the formulation was essential to lower the batter density to a more desirable value, closing to that of the standard (100% WF). Ovalette appeared to enhance the aeration and help stabilize the foams in the batter. Ovalette is a commercial emulsifier consisting of a mixture of α-tending emulsifiers (mono- and diglycerides and polyglycerol esters of fatty acids) that is commonly used in cake production. Blends of monoglycerides and polyglycerol esters of fatty acids bring about synergistic effect in enhancing the formation of firm and stable gels via attractive integrations and forming "bridges" between air bubbles in foams that lead to more solid-like foams [16,44] to effectively retain the entrapped air. Blends of polyglycerol fatty acid esters and monoglycerides are known to improve sponge cake aeration and stability with less mixing time and improved foam and emulsion stability [16,45].

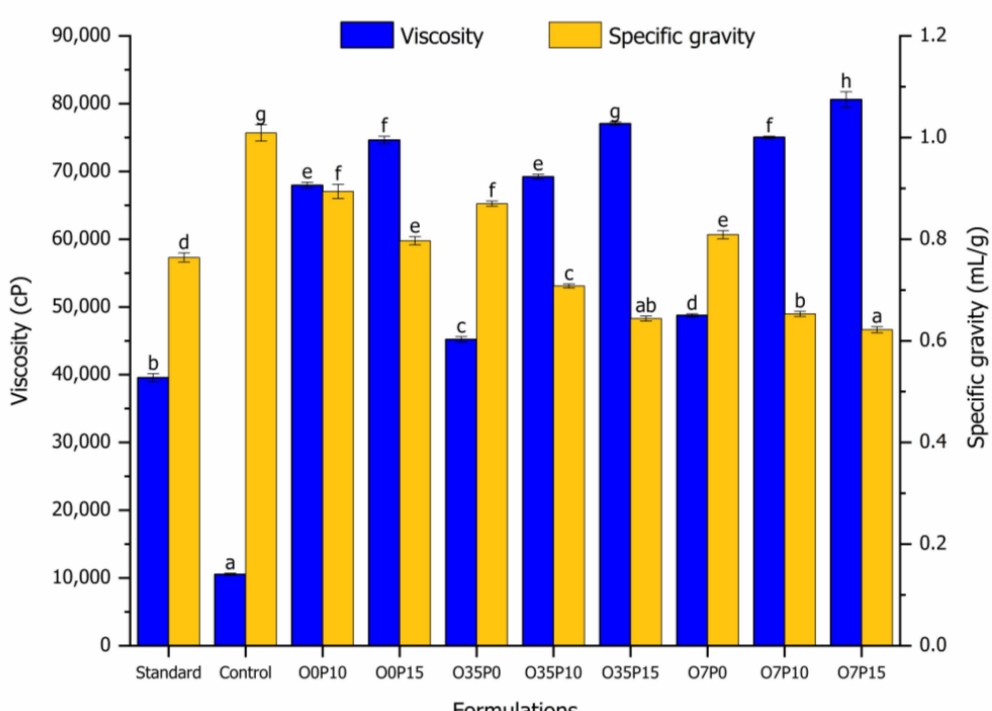

**Figure 1.** Viscosity (cP) and specific gravity (mL/g) for different formulations of cake batter. Histogram with different alphabet (a–h) indicates significant difference ($p < 0.05$).

*3.3. Cake Analysis*

3.3.1. Specific Volume and Weight Loss

Specific volume and weight loss were determined for the cakes (Figure 2). Specific volume is the indicator of the strength and extensibility of the food matrix, it is also a critical visual quality for the cake. Weight loss during the cooking of the batter is related to the gas escaping during steaming and it is a crucial parameter for the structural transformation of the cake [39]. Formation of cake structure primarily relies on the aeration and gas bubble stability during cooking [46]. The control sample presented a very compact structure with the lowest degree of expansion (Figure S1). Because of its extremely low batter viscosity (Figure 1), this sample failed to hold the gas bubbles within the food matrix while steaming. The results obtained suggest that the presence of both additives are important for the improvement of the cake volume; though the specific volume of the sample added with SPI alone (O0P10) achieved a similar specific volume to the standard ($p < 0.05$), but the weight loss was still far higher than the desirable value. When SPI and Ovalette were incorporated in the formulation at an appropriate ratio, O35P15, the specific volume and weight loss were insignificantly different from the standard ($p > 0.05$). The batter viscosity increments by addition of SPI and Ovalette (Figure 1) created a sufficient batter consistency that is crucial for retaining the air bubble formed during mixing and the $CO_2$ produced during steaming. However, the progressive addition of SPI was unfavorable because the batter density turned out to be too high to effectively incorporate the desirable amount of gas bubbles. As shown in Figure 2, supplementing 10% of SPI showed enhancement in the cake volume, but when 15% of SPI was used, the volume of the cakes was impaired instead. Apparently, the correct batter viscosity is crucial to ensure the successful aeration of the cake. Majzoobi et al. [29] found that an acceptable sponge cake can be obtained by using 20% SPI, but any further increase in SPI resulted in inferior cake quality. When used in bread batter, emulsifiers promoted the viscosity, which in turn improved the buoyancy of bubbles, preventing their coalescence and producing a uniform distribution [46]. The $\alpha$-gel phase of the emulsifier would form solid and elastic films at the oil–water interface, which encapsulated oil during air incorporation into cake batter, thus preventing foam destabilization [18,47].

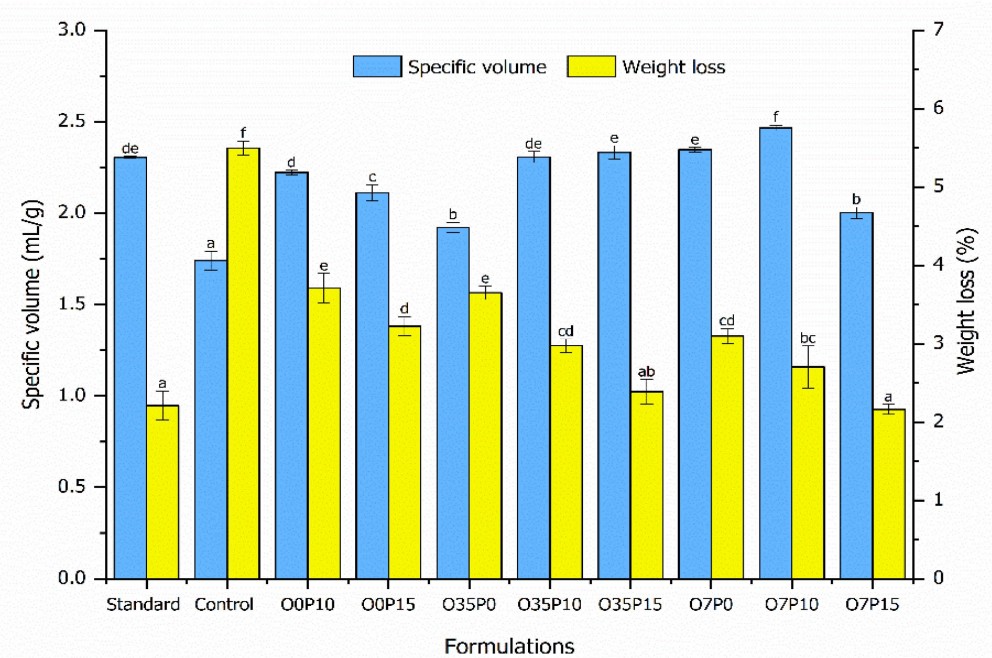

**Figure 2.** Specific volume (mL/g) and weight loss (%) of different formulations of Saba banana flour cake. Histogram with different alphabet (a–f) indicates significant difference ($p < 0.05$).

3.3.2. Texture Profile Analysis

Apart from visual quality, the texture of the cake is another critical characteristic that may influence consumer acceptability. The hardness, cohesiveness, springiness and chewiness of the cakes were determined (Table 4). As expected, those cakes with a less expanded volume (control, O0P10, O0P15 and O7P15) (Figure S1) recorded higher hardness than the standard ($p < 0.05$). The highly compact stacking of starch particles in these samples required higher external force to cause the structural damage. Even though addition of SPI was able to increase the specific volume in O0P10 and O0P15 (Figure 2), it did not improve the hardness of the cakes ($p < 0.05$); rather, the hardness of the cake increased with the increasing level of SPI in these two samples which may be attributed to the high water binding capacity of SPI that reduced the free water and hence the cake softness [29]. On the other hand, the hardness of the sample supplemented with Ovalette alone (O35P0 and O7P0) showed an insignificant difference from the standard ($p > 0.05$). Results obtained also indicate that Ovalette inclusion softened the banana flour cakes added with SPI such as O35P10, O35P15 and O7P10. The softening effect of Ovalette was credited to the formation of amylose-emulsifier complex that reduced the recrystallization of starch molecules after cooking and upon cooling [19]. These complex structures prevent further physical changes in dissolved amylose and reducing starch retrogradation [18]. The extremely high hardness of O7P15 could be attributed to the highest solid content and relatively low specific volume (dense structure).

**Table 4.** Texture profile of wheat flour cake (standard) and different formulations of Saba banana flour cakes.

| Sample | Texture Profile Parameters | | | |
| --- | --- | --- | --- | --- |
| | Hardness (kg) | Cohesiveness | Springiness | Chewiness (kg) |
| Standard | 2.517 ± 0.129 [a] | 0.674 ± 0.018 [f] | 0.880 ± 0.009 [c] | 1.493 ± 0.086 [d] |
| Control | 3.288 ± 0.224 [c] | 0.400 ± 0.014 [e] | 0.715 ± 0.163 [b] | 0.943 ± 0.240 [c] |
| O0P10 | 3.454 ± 0.211 [c] | 0.380 ± 0.022 [d] | 0.689 ± 0.137 [ab] | 0.898 ± 0.162 [bc] |
| O0P15 | 3.784 ± 0.177 [d] | 0.374 ± 0.011 [d] | 0.666 ± 0.151 [ab] | 0.944 ± 0.228 [c] |
| O35P0 | 2.473 ± 0.159 [a] | 0.383 ± 0.018 [d] | 0.740 ± 0.070 [b] | 0.702 ± 0.099 [a] |
| O35P10 | 2.605 ± 0.188 [a] | 0.371 ± 0.008 [d] | 0.665 ± 0.061 [ab] | 0.629 ± 0.075 [a] |
| O35P15 | 2.768 ± 0.167 [ab] | 0.354 ± 0.015 [c] | 0.591 ± 0.116 [a] | 0.596 ± 0.102 [a] |
| O7P0 | 2.405 ± 0.140 [a] | 0.369 ± 0.009 [cd] | 0.726 ± 0.072 [b] | 0.646 ± 0.092 [a] |
| O7P10 | 2.875 ± 0.231 [b] | 0.335 ± 0.012 [b] | 0.687 ± 0.135 [ab] | 0.655 ± 0.116 [a] |
| O7P15 | 3.674 ± 0.196 [d] | 0.309 ± 0.008 [a] | 0.663 ± 0.112 [ab] | 0.751 ± 0.128 [ab] |

Means ($n = 3$) in a column with identical alphabet indicate insignificant difference ($p > 0.05$). Standard—cake made of WF; control—cake made of GSBF without SPI and Ovalette.

The results obtained indicating that gluten is the main factor granting the typical cohesiveness, springiness and chewiness of the cake. Cohesivity of gluten is known for providing necessary structuring functionalities in bakery products. In the absence of gluten, these textural properties were totally diminished ($p < 0.05$) as shown in the textural parameters of the control. Cohesiveness represents the ability of a material to stick to itself; thus, it is an indication of how well a product withstands a second deformation relative to its resistance under the first deformation [48]. It also refers to the rate at which food disintegrates under mechanical action, or as the resistance of food to traction [29]. Increasing the level of SPI and Ovalette was found to further reduce the cohesiveness of the GSBF cakes ($p < 0.05$). High springiness values are related to high chewing quality, whereas low springiness reflects the tendency to crumble upon external forces [49–51]. Comparatively, this textural parameter seemed to be least influenced by SPI and Ovalette where almost all the cake formulations showed similar springiness to the control ($p > 0.05$). Chewiness is defined as the energy required to masticate a solid food product [52]. Without Ovalette, the chewiness of O0P10 and O0P15 were insignificantly different to the control, but other cake samples supplemented with both Ovalette and SPI required much lower

energy to chew with no difference among them ($p < 0.05$). The use of SPI and Ovalette was incapable of improving the cohesiveness and chewiness of GSBF cake, instead causing an appreciable reduction in these parameters.

### 3.3.3. Sensory Evaluation

Ranking Test was conducted to screen the eight formulations of GSBF-based cake formulations supplemented with SPI and Ovalette, and the rank sum obtained for each sample is shown in Table 5. Comparatively, the formulations without SPI (O35P0 and O7P0) were least preferred by the panelists. The four top-ranked samples with insignificant statistical difference ($p > 0.05$) were identified as O35P10, O35P15, O7P10 and O0P10. These four samples were then subjected to the Hedonic Test to compare the degree of satisfaction and acceptance regarding the cake appearance, color, aroma, taste, softness, moistness and overall acceptability. The control was deliberately included in the Hedonic Test to be compared with these four samples.

**Table 5.** Rank sum for green Saba banana flour-based cakes obtained in Ranking Test.

| Sample | Rank Sum |
|---|---|
| O35P10 | 36 [a] |
| O35P15 | 39 [a] |
| O7P10 | 40 [ab] |
| O0P10 | 45 [ab] |
| O7P15 | 56 [c] |
| O0P15 | 64 [cd] |
| O35P0 | 70 [cd] |
| O7P0 | 72 [d] |

Rank sums with identical alphabet indicates insignificant difference ($p > 0.05$). *LSD* rank = 15.18.

Table 6 shows the mean scores for the seven tested sensory attributes. In agreement with the results of the cake analysis and TPA, the control obtained the lowest degree of satisfaction for all the tested attributes ($p < 0.05$). In brief, the control was disliked by the panelists, whereby the mean scores for all the attributes fell between 2—dislike very much and 3—dislike moderately. The results of the Hedonic Test show that the poor eating quality of the control was effectively overcome by the addition of SPI and Ovalette. It is notable that the cakes supplemented with both additives (O35P10, O35P15, O7P10) were more preferred to the one that received SPI alone (O0P10). An appreciable increase in preference was observed in all the supplemented samples except for color. This is because the color of the cakes was affected by the darker color of the green banana flour (Table 3), the crumbs turned out to be brownish, differing from the golden yellowish of normal cake (Figure S1). Cakes with a higher volume exhibited a higher preference, possibly due to the brighter color as affected by the degree of expansion. Sample O35P15 was the most preferred sample that obtained the highest score for all the tested attributes ($p < 0.05$), hence was rated with the highest overall acceptability. It should, however, be borne in mind that the sensory quality for O35P15 may require further enhancement in future as the mean score for overall acceptability only lay between 7—like moderately and 8—like very much.

### 3.4. Chemical Composition of Selected Cake Formulations

The effect of SPI and Ovalette on the macronutrient content of GSBF cake in the most preferred formulation (O35P15) was determined by comparing it to the composition of the standard and the control (Table 7).

**Table 6.** Mean score for sensory attributes of five formulations of Saba banana flour cake obtained from Hedonic Test.

| Attribute | Sample | | | | |
|---|---|---|---|---|---|
| | Control | O0P10 | O35P10 | O35P15 | O7P10 |
| Cake Volume | $2.43 \pm 0.78$ [a] | $4.11 \pm 0.27$ [b] | $6.73 \pm 0.86$ [c] | $7.38 \pm 0.29$ [c] | $6.83 \pm 0.55$ [c] |
| Color | $2.78 \pm 0.55$ [a] | $3.38 \pm 0.44$ [b] | $5.10 \pm 0.29$ [c] | $5.40 \pm 0.38$ [d] | $5.38 \pm 0.47$ [d] |
| Aroma | $2.68 \pm 1.03$ [a] | $6.73 \pm 0.29$ [b] | $7.15 \pm 0.61$ [bc] | $7.83 \pm 0.35$ [c] | $6.98 \pm 0.33$ [b] |
| Taste | $2.55 \pm 1.32$ [a] | $6.00 \pm 0.36$ [b] | $7.02 \pm 0.09$ [c] | $7.65 \pm 0.12$ [d] | $6.70 \pm 0.45$ [c] |
| Softness | $2.16 \pm 0.41$ [a] | $6.48 \pm 0.20$ [b] | $7.38 \pm 0.59$ [d] | $7.75 \pm 0.11$ [d] | $7.11 \pm 0.88$ [c] |
| Moistness | $2.88 \pm 0.39$ [a] | $6.20 \pm 0.61$ b | $7.78 \pm 0.58$ [d] | $7.58 \pm 0.43$ [d] | $7.25 \pm 0.63$ [c] |
| Overall Acceptability | $2.48 \pm 0.65$ [a] | $5.38 \pm 0.72$ [b] | $6.55 \pm 0.27$ [c] | $7.30 \pm 0.31$ [d] | $6.35 \pm 0.41$ [c] |

Means ($n = 40$) in a row with identical alphabet indicate insignificant difference ($p > 0.05$). Control—cake made of GSBF without SPI and Ovalette. Descriptor for 9-point Hedonic Scale: 1—dislike extremely; 5—neither like nor dislike; 9—like extremely.

**Table 7.** Chemical composition of standard (100% WF), control (100% SGBF) and O35P15.

| Composition | Standard | Control | O35P15 |
|---|---|---|---|
| Moisture (%) | $33.74 \pm 0.24$ [b] | $31.44 \pm 0.16$ [a] | $31.58 \pm 0.15$ [a] |
| Protein * (%) | $10.79 \pm 0.04$ [b] | $7.16 \pm 0.03$ [a] | $12.67 \pm 0.06$ [c] |
| Fat * (%) | $10.70 \pm 0.25$ [c] | $8.32 \pm 0.02$ [a] | $10.13 \pm 0.17$ [b] |
| Ash * (%) | $1.53 \pm 0.11$ [a] | $1.69 \pm 0.09$ [a] | $1.84 \pm 0.10$ [b] |
| Dietary fiber * (%) | $3.57 \pm 0.09$ [a] | $13.66 \pm 0.17$ [c] | $12.70 \pm 0.11$ [b] |
| Resistant starch * (%) | $2.10 \pm 0.17$ [a] | $13.02 \pm 0.35$ [c] | $8.51 \pm 0.34$ [b] |

* Dry matter basis. Means with identical alphabet within the same row indicate insignificant difference ($p > 0.05$). Standard—cake made of WF; control—cake made of GSBF without SPI and Ovalette.

The nutrient profile, particularly protein and resistant starch content, of O35P15 was significantly improved as compared to the control ($p < 0.05$). SPI was reported to contain 8.41% protein [17], thus resulted in an almost 77% increment in protein when compared to the control. The protein content in O35P15 is even higher than that of the standard ($p < 0.05$). The high percentage of SPI in O35P15 is also believed to contribute to the higher fat content ($p < 0.05$), in which the SPI used in this study was reported to contain 5% of crude fat (acid hydrolysis). It is remarkable that the dietary fiber and resistant starch in both of the banana flour-based cakes were much higher than the standard ($p < 0.05$), owing to the nutrient profile of SGBF (Table 1). The RS in banana cakes was much lower than that of GSBF possibly ascribed to the presence of different types of RS in the materials. RS is classified into five subtypes, RS1 (physically inaccessible starch), RS2 (raw starch with B-type crystalline), RS3 (retrograded starch), RS4 (chemically modified starch) and RS5 (amylose-lipid complexes) [52,53]. The RS in GSBF is primarily comprised of RS2 the indigestibility of which is inherently due to the more perfectly arranged crystalline molecular structure [40]. Steaming caused disruption of the starch granules in the batter and destroyed the majority of RS2 in raw GSBF to increase the starch digestibility. Upon cooling of the cake, part of the gelatinized starch experienced retrogradation to yield RS3. These newly rearranged crystals were resistant to the enzymatic attack during digestion. Retrogradation of amylose molecule was identified as the main mechanism for the formation of RS3 but banana starch also contained long outer amylopectin $\alpha$-1,6-linked side chains that were an excellent source for producing RS3 [54,55]. According to Codex Alimentarius International Food Standards [56], the food is considered a source of dietary fiber if the dietary fiber content is at least 3 g of fiber per 100 g of food, and high in fiber if it contains 6 g of fiber per 100 g of food. Equally, a food that contains 10 g of protein per 100 g of food can be considered as high in protein. With regards to this, O35P15 can be claimed as a food high in protein and high in dietary fiber.

## 4. Conclusions

This study demonstrated the feasibility of employing SPI and Ovalette in overcoming the technological drawbacks of SGBF in cake making. These two additives significantly

improved the technological characteristics of the batter and hence the cake. Noticeable enhancement of nutritional quality was also observed in the best formulated GSBF cake produced, particularly in the protein, dietary fiber and resistant starch contents. This finding confirmed the potential of green Saba banana flour to be included as an ingredient to rectify the low dietary fiber intake in human diet. However, the results of TPA and sensory evaluation suggest that the texture and color of the cake did not achieve the expected level of acceptance. Based on the results of the Hedonic Test, the overall acceptability of the best cake formulation was rated in between the score "like moderately" and the score "like very much", therefore, future work looking into further improvement of the texture and color of the cake is recommended.

**Supplementary Materials:** The following supporting information can be downloaded at: https://www.mdpi.com/article/10.3390/app13042421/s1, Figure S1: Cross sectional view of standard (100% wheat flour), the control (100% GSBF) and other GSBF cakes supplemented with different levels of soy protein isolate (SPI) and Ovalette; Table S1: The plan for arranging GSBF cakes for Ranking Test generated by Balanced Incomplete Block (BIB) Design.

**Author Contributions:** Conceptualization, J.-S.L.; methodology, A.L.H.; validation, J.-S.L. and N.Y.; formal analysis, W.X.T.; investigation, W.X.T.; resources, J.H.A. and N.Y.; data curation, J.-S.L. and W.X.T.; writing—original draft preparation, J.-S.L. and C.K.S.; writing—review and editing, J.-S.L. and A.L.H.; visualization, W.X.T. and J.H.A.; supervision, J.-S.L.; project administration, N.Y. and C.K.S.; funding acquisition, J.-S.L. All authors have read and agreed to the published version of the manuscript.

**Funding:** This research was funded by Universiti Malaysia Sabah under Niche Research Grant Scheme (SDN0038-2019).

**Institutional Review Board Statement:** Not applicable.

**Informed Consent Statement:** Informed consent was obtained from all sensory panelists involved in the study.

**Data Availability Statement:** Not applicable.

**Conflicts of Interest:** The authors declare no conflict of interest.

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
