# Peer review of "Quality Improvement of Green Saba Banana Flour Steamed Cake"

_applsci, doi:10.3390/app13042421_

Round 1

Reviewer 1 Report

Comments for Authors,

The manuscript entitled "Quality improvement of green Saba banana flour steamed cake " is suitable for publication in Applied Sciences; however, it has to be improved in some aspects, considering the following specific comments.

Specific Comments:

Line 148 and 149, The authors should indicate (in parentheses), the value of the ambient temperature.

Line 156, Should read as: … Five g of flour …

Line 161, Should read as: One g of flour …

Line 163, Should read as: … at 3,000 ´ g for …

Line 172, Should read as: One g of flour …

Line 172, Palm? Is this correct, or are the authors referring to palm oil? Please clarify this!

Line 174, Should read as: 3,000 ´ g for …

Line 211, Should read as: Four mL of …

Line 213, Should read as: Six mL of …

Line 224, In formula no. 3 (Resistant starch), the authors should indicate: What does the number 90 refer to?

Line 237, Should read as: Two hundred of …

Line 239, Should read as: … after 1 min …

Line 247, Should read as: Thirty s delay …

Line 264, Should read as: Forty healthy panelists …

In relation to the footnotes of Tables 2, 3, 5 and 6, the authors indicate: Means with identical alphabet within the same column …; however, I notice that the comparison is done based on the rows rather than the columns. Please check this out!

Lines 321 and 412: Both are indicating Table 3. Please correct this!

Author Response

Please refer attached file.

Reviewer 2 Report

Corrections have been marked in the PDF. 

Author Response

Please refer attached file.

Reviewer 3 Report

The publication is interesting and concerns the improvement of the quality of gluten-free steamed bread from green Saba banana flour through the use of soy protein isolate and commercial emulsifiers.

However, I have a few remarks:

Table 1 should be corrected, wrong amount of Ovalette and SPI in the recipe. In the abstract there is information that the Ovalette supplement was 0, 3.5 and 7% and the SPI 0, 10 and 15% and the table shows other values.

There is an error in the description of the last attempt (O7910), instead of P there is 9.

Oil holding capacity: The description of the method lacks the step of mixing flour with palm oil.

Line 172: The word oil for palm is missing.

Line 209: It should be amyloglucosidase and it is amylogluosidase.

Line 268: There is talk of a 9-point hedonic scale and in parentheses the maximum score mentioned is 7.

Table 2: I don't understand the designation of homogeneous groups in the table at all. The table description says: Means with identical alphabet within the same column indicate insignificant difference, but I can't agree with that. For wheat flour (WF) the values of 13.44% and 0.91% are in the same homogeneous group. And 0.91% and 0.85% are already in other homogeneous groups. If we want to compare WF with GSBF, we should compare average values in rows, not columns.

Line 306: it should be -hence higher RS than in WF.

Table 3: I don't understand the designation of homogeneous groups in the table again. Even if we were to compare the values in the rows, there is no rule that the higher value is marked with the letter "a" and the lower value "b" or vice versa. These markings seem to be given randomly.

Line 366: I believe it should be in the sentence - "control sample" not just "control". A shortcut was used.

Line 396: I cannot agree that the O35P0 sample was characterized by a higher hardness than the standard, because they are in the same homogeneous group. The samples with the addition of SPI without the addition of Ovalette, i.e. O0P10 and O0P15, are characterized by higher hardness than the standard, values above 3.0 kg.

Line 433: I cannot agree with the opinion regarding Chewiness. The control and sample O0P15 are in the same homogeneous group.

Line 439: In my estimation, the least preferred samples were those without SPI, not the Ovalette -O35P0 and O7P0.

Table 5, 6: Again, an error in the description of homogeneous groups - not in a column with identical alphabet indicate insignificant difference but in a row.
